# The Inhibitory Roles of Vitamin K in Progression of Vascular Calcification

**DOI:** 10.3390/nu12020583

**Published:** 2020-02-23

**Authors:** Atsushi Shioi, Tomoaki Morioka, Tetsuo Shoji, Masanori Emoto

**Affiliations:** 1Department of Vascular Medicine and Vascular Science Center for Translational Research, Osaka City University Graduate School of Medicine, Osaka 545-8585, Japan; t-shoji@med.osaka-cu.ac.jp; 2Department of Metabolism, Endocrinology, and Molecular Medicine, Osaka City University Graduate School of Medicine, Osaka 545-85858, Japan; m-tomo@med.osaka-cu.ac.jp (T.M.); memoto@med.osaka-cu.ac.jp (M.E.)

**Keywords:** atherosclerosis, matrix Gla protein, oncostatin M, vascular calcification

## Abstract

Vitamin K is a fat-soluble vitamin that is indispensable for the activation of vitamin K-dependent proteins (VKDPs) and may be implicated in cardiovascular disease (CVD). Vascular calcification is intimately associated with CV events and mortality and is a chronic inflammatory process in which activated macrophages promote osteoblastic differentiation of vascular smooth muscle cells (VSMCs) through the production of proinflammatory cytokines such as IL-1β, IL-6, TNF-α, and oncostatin M (OSM) in both intimal and medial layers of arterial walls. This process may be mainly mediated through NF-κB signaling pathway. Vitamin K has been demonstrated to exert anti-inflammatory effects through antagonizing NF-κB signaling in both in vitro and in vivo studies, suggesting that vitamin K may prevent vascular calcification via anti-inflammatory mechanisms. Matrix Gla protein (MGP) is a major inhibitor of soft tissue calcification and contributes to preventing both intimal and medial vascular calcification. Vitamin K may also inhibit progression of vascular calcification by enhancing the activity of MGP through facilitating its γ-carboxylation. In support of this hypothesis, the procalcific effects of warfarin, an antagonist of vitamin K, on arterial calcification have been demonstrated in several clinical studies. Among the inactive MGP forms, dephospho-uncarboxylated MGP (dp-ucMGP) may be regarded as the most useful biomarker of not only vitamin K deficiency, but also vascular calcification and CVD. There have been several studies showing the association of circulating levels of dp-ucMGP with vitamin K intake, vascular calcification, mortality, and CVD. However, additional larger prospective studies including randomized controlled trials are necessary to confirm the beneficial effects of vitamin K supplementation on CV health.

## 1. Introduction

Vitamin K is a fat-soluble vitamin that is composed of a 2-methyl-1,4-naphthoquinone ring and a side chain at the 3-carbon position varying in length and degree of saturation [1]. Vitamin K naturally occurs in two forms: vitamin K1 (phylloquinone) and vitamin K2 (menaquinones (MKs)). Vitamin K1 is predominantly found in green leafy vegetables and plant oils such as soybean, canola, and olive, whereas vitamin K2 is present in small amounts in fermented foods, milk products, meat, and cheese. Vitamin K2 is the most potent form and has a longer half-life compared with vitamin K1 [2]. Moreover, MK-4 is the most prevalent form of vitamin K in human and animal tissues and can be formed from menadione through the action of UbiA prenyltranferase domain-containing protein 1 (UBIAD1) [3]. 

The most important function of vitamin K is to serve as a cofactor in the synthesis of vitamin K-dependent coagulation factors II, VII, IX, and X in the liver [4,5]. Activation of vitamin K-dependent proteins (VKDPs) is mediated through the conversion of glutamic acid (Glu) residue of their molecules to γ-carboxyglutamic acid (Gla) by the action of γ-glutamyl carboxylase (GGCX). This process proceeds by the oxidation of vitamin K hydroquinone (KH_2_) to vitamin K epoxide (KO) in the vitamin K cycle. KO is reduced to KH2 by vitamin K epoxide reductase (VKOR). Vitamin K antagonists such as warfarin exert their anti-coagulative effects by inhibiting VKOR activities. 

Two natural anticoagulants, protein C and protein S, are the vitamin K-dependent plasma proteins that regulate blood coagulation by inhibiting activated factors Va and VIIIa (FVa and FVIIIa) [6]. In addition to its anticoagulant activity, activated protein C (APC) exhibits anti-inflammatory and anti-apoptotic effects by its binding to endothelial cell protein C receptor (EPCR), leading to activation of protease-activated receptor 1 (PAR1) [7]. Protein S functions as a cofactor for APC in the degradation of FVa and FVIIIa. Protein S also exerts anti-inflammatory effect through its binding to the Tyro3/Axl/Mer (TAM) family of receptor tyrosine kinases [8].

Other extra-hepatic VKDPs such as matrix Gla protein (MGP), osteocalcin, and Gas6, and Gla-rich protein (GRP) have been identified. Osteocalcin (also called bone Gla protein) is mainly secreted by osteoblasts and facilitates the deposition of calcium into bone matrix [9,10]. Matrix Gla protein (MGP) is predominantly produced by chondrocytes and vascular smooth muscle cells and functions as a potent calcification inhibitor within the arterial walls [10,11]. Growth arrest-specific 6 (Gas6) is known to be a ligand for the TAM family of receptor tyrosine kinases and prevents endothelial cells and vascular smooth muscle cells (VSMCs) from undergoing apoptosis and inhibits vascular calcification through its anti-apoptotic effect on VSMCs [10,12]. Gla-rich protein (GRP) was identified in sturgeon cartilage as a novel VKDP containing many Gla residues in the molecule (16 Gla residues among 74 amino acids) and may serve as an inhibitor of vascular calcification [10,13]. 

Vitamin K has been shown to function as an anti-inflammatory factor, independent of its activity as a cofactor for GGCX. Vitamin K status was inversely associated with circulating inflammatory markers such as IL-6 and C-reactive protein (CRP) [14]. Vitamin K attenuated lipopolysaccharide (LPS)-induced inflammatory responses by blocking nuclear factor κB (NF-κB) signal transduction [15,16,17]. 

Cardiovascular disease (CVD) is the leading cause of morbidity and mortality all over the world. Vascular calcification is a striking feature of chronic inflammatory diseases such as atherosclerosis, type 2 diabetes, and chronic kidney disease (CKD) and is associated with increased risk of adverse CV events [18,19,20,21]. Vitamin K has been proposed to show a protective effect against CVD through the action of VKDPs such as MGP to inhibit vascular calcification [22,23]. Furthermore, anti-inflammatory effects of vitamin K may be involved in preventing progression of atherosclerotic plaque calcification. In this review, we will describe the inhibitory roles of vitamin K and MGP in vascular calcification and the clinical significance of inactive MGP (dephospho-uncarboxylated MGP: dp-ucMGP) as a novel biomarker of vascular calcification and CVD. 

## 2. Vascular Calcification and CVD 

Vascular calcification is a hallmark of atherosclerosis especially in coronary arteries. Particularly in patients with diabetes mellitus and CKD, coronary artery calcification (CAC) is strikingly accelerated and predicts future CV events and all-cause mortality [19,24]. 

Morphologically, two types of vascular calcification have been described. Intimal calcification usually develops in accordance with progression of atherosclerosis and may cause coronary ischemic events. On the other hand, medial calcification is independent of atherosclerosis and predominantly develops along elastic fibers. Consequently, medial calcification promotes arterial stiffness and increases pulse pressure as well as systolic blood pressure, resulting in left ventricular hypertrophy, diastolic dysfunction, and heart failure [25]. 

VSMCs play a pivotal role in vascular calcification. Under procalcifying conditions, VSMCs undergo osteoblastic differentiation and express Runx2, the master transcription factor for osteogenesis along with other bone-related proteins such as alkaline phosphatase (ALP) and bone sialoprotein II (BSP-II) [26]. These transdifferentiated osteoblast-like cells generate matrix vesicles and exosomes that initiate the mineralization process and deposit bone-like matrix within the arterial wall. These processes are promoted by loss of calcification inhibitors, oxidative stress, endoplasmic reticulum stress, apoptosis, and DNA damage response signaling [26]. 

## 3. Inflammation and Vascular Calcification

Vascular calcification is an inflammation-mediated process where macrophages promote osteoblastic differentiation and mineralization of VSMCs via secreting proinflammatory cytokines such as interleukin (IL)-1β, IL-6, tumor necrosis factor-α (TNF-α), and oncostatin M (OSM) [27,28,29,30]. In intimal vascular calcification, plaque inflammation evoked by infiltrated macrophages precedes active calcification [31,32], while an increased number of CD68-positive macrophages are localized in the vascular wall associated with medial calcification in CKD [33,34]. Moreover, it has been suggested that pro-inflammatory cytokines such as TNF-α may play an important role in the development of medial calcification associated with diabetes and CKD [35]. 

It has been shown that NF-κB activation by TNF-α and LPS promotes osteogenic differentiation of human adipose tissue-derived mesenchymal stromal/stem cells [36,37]. Moreover, TNF-α also enhanced osteogenic differentiation of both human bone marrow-derived mesenchymal stem cells and human dental pulp stem cells through activation of NF-κB signaling pathway [38,39]. Furthermore, NF-κB signaling stimulated by various stimuli such as high glucose, high phosphate, and oxidative stress as well as proinflammatory cytokines has also been involved in osteogenic differentiation of VSMCs [40,41,42,43,44]. Therefore, NF-κB signaling play a pivotal role in osteogenic differentiation of not only mesenchymal stem cells, but also VSMCs.

## 4. Anti-Inflammatory Effects of Vitamin K

Anti-inflammatory action of vitamin K has been shown in several in vitro and in vivo studies. MK-4 suppressed LPS-induced expression of inflammatory cytokines such as IL-6 in macrophage-like cells and MG-6 mouse microglia-derived cells through the inhibition of NF-κB signaling pathway [16,45]. Pretreatment with MK-7 inhibited the production of TNF-α after the toll-like receptor (TLR) activation in human monocyte-derived macrophages [17]. LPS-induced production of IL-6 in human fibroblasts is intensively inhibited by naphthoquinone compounds [46]. In an in vivo study utilizing rats, vitamin K1 supplementation suppresses the inflammation induced by LPS as evidenced by plasma levels of transaminases and hepatic mRNA levels of macrophage migration inhibitory factor [15]. Vitamin K1 also attenuates streptozotocin-induced diabetes in rats by reducing free radical stress and suppressing NF-κB activation [47]. Clinical studies have showed that vitamin K status is negatively associated with circulating levels of inflammatory markers. Plasma K1 levels were inversely associated with IL-6 and C-reactive protein (CRP) in a cross-sectional study conducted with 379 healthy men and women [48]. However, no significant changes of the levels of inflammatory biomarkers were observed in the 3-year follow-up of patients supplemented with vitamin K1 [48]. In another cross-sectional analysis of the Framingham Offspring Study (*n* = 1381), plasma levels of phylloquinone (vitamin K1) and its intake were inversely associated with overall circulating markers of inflammation, including CD40 ligand and IL-6 [14]. Higher serum phylloquinone levels were associated with several serum inflammatory markers such as IL-6, soluble intercellular adhesion molecule-1 (ICAM-1), and CRP in a cross-sectional study conducted with 662 community-dwelling adults from the Multi-Ethnic Study of Atherosclerosis (MESA) [49]. Therefore, it is likely that vitamin K may prevent inflammatory vascular diseases including atherosclerosis and vascular calcification through its anti-inflammatory actions on vascular cells. 

## 5. Matrix Gla Protein and other VKDPs in Vascular Calcification

Matrix Gla protein (MGP) is an 84-amino acid VKDP that is secreted by chondrocytes and VSMCs and is expressed in not only bone, but also the heart, vessels, kidneys, and cartilage. MGP contains five glutamic acid residues and three serine residues and this molecule is activated through two post-translational modifications as follows: vitamin K-dependent carboxylation of glutamate and serine phosphorylation [50]. After these maturation steps, activated MGP serves as a potent calcification inhibitor. 

Mice lacking MGP developed extensive arterial calcification and die within six to eight weeks due to rupture of the aorta [51]. Histological analyses clarified extensive calcification along with elastic fibers and the presence of chondrocyte-like cells in the media. Interestingly, restoration of MGP expression in VSMCs rescued the arterial calcification phenotype in MGP-deficient mice, while raising circulating levels of MGP through overexpression of MGP transgene in the liver did not affect vascular calcification, suggesting that only expression of MGP in VSMCs can inhibit vascular calcification [52]. MGP expression in VSMCs was inversely regulated by the development of VSMC calcification [53]. In humans, mutations in the gene encoding MGP causes Keutel syndrome, a rare autosomal recessive disorder characterized by severe calcification of soft tissues [54,55]. The possible mechanisms have been suggested for the inhibitory actions of MGP on calcification. Firstly, MGP binds hydroxyapatite crystals thereby inhibiting crystal growth [56]. Secondly, MGP has the capacity to bind bone morphogenetic protein-2 (BMP-2) and to attenuate its activity [57,58]. BMP-2 promotes transdifferentiation of VSMCs into osteoblast-like cells. MGP has been shown to be localized in not only calcified atherosclerotic plaques, but also medial calcified lesions [59,60,61,62,63]. Thus, MGP has been recognized as a major inhibitor of both intimal and medial vascular calcification. 

As mentioned in the previous section, VKDPs other than MGP may also function as inhibitors of vascular calcification. Gas6 exerts its inhibitory effect on vascular calcification through its binding to the Axl receptor. Interaction between Gas6 and Axl activates the PI3K/Akt pathway resulting in stimulation of cell survival pathway [64]. Therefore, apoptosis-mediated VSMC calcification can be inhibited by the Gas6/Axl-PI3K/Akt pathway activated by statin and vitamin K2 [65,66]. GRP also function as an inhibitor of vascular calcification. GRP is immunohistochemically localized at the site of mineral deposition in human aorta and aortic valve tissues [67]. The gene expression of GRP is also significantly upregulated in calcified aortic valve tissues compared with non-calcified aortic valve tissues [68]. VSMCs derived from GRP-deficient mice exhibit an increased capacity of in vitro calcification and expression of osteogenic markers such as Runx2, ALP, and OCN [68]. Protein C and protein S deficiency may be involved in the pathogenesis of calcific uremic arteriolopathy (also known as calciphylaxis) characterized by skin ulcer and tissue necrosis [69].

## 6. Vitamin K and CAC

CAC is closely related to the overall plaque burden of coronary artery atherosclerosis. Higher CAC score is associated with a greater risk of adverse CV events and all-cause mortality [70,71,72]. A preventive role for vitamin K against CAC progression has been proposed through the inhibitory action of MGP on mineral deposition within the arterial wall [22,73]. Once MGP is activated through vitamin K-dependent γ-carboxylation and subsequent serine phosphorylation, it potently suppresses arterial calcification [50]. Anti-inflammatory effects of vitamin K, independent of γ-carboxylation, may also contribute to the inhibition of CAC as mentioned above. 

### 6.1. Vitamin K Status and CAC

There have been several observational studies assessing the relationship between vitamin K status and CAC. Vitamin K status has been assessed by vitamin K intake, circulating levels of phylloquinone (PK) or menaquinone (MK), and measuring uncarboxylated fractions of certain VKDPs such as MGP [2,74]. In an observational study among U.S. military personnel at low CVD risk, phylloquinone intake was not associated with CAC [75]. In a case-cohort study conducted on MESA subjects, low levels of serum vitamin K1 were significantly associated with CAC progression in patients treated with antihypertensive drugs [76]. In a cross-sectional study of 564 post-menopausal women to examine the association of PK and MK intake with CAC, PK intake was not associated with CAC [RR (95% CI):1.17 (0.96–1,42); the highest versus lowest quartile], while MK intake was associated with reduced CAC [RR (95% CI):0.80 (0.65–0.98); the highest versus lowest quartile] [77]. In an observational study to investigate the association of CAC and circulating levels of PK, MK-4, and MK-7, there was no significant correlation of CAC with plasma levels of PK, MK-4, and MK-7 [78]. In another cross-sectional study among 40 hemodialysis (HD) patients, circulating uc-MGP levels as a vitamin K status marker were inversely associated with the severity of CAC [79].

### 6.2. Vitamin K Supplementation and CAC

There are a few studies examining the effect of vitamin K supplementation on coronary artery calcification. A 3-year prospective randomized controlled trial (RCT) revealed that PK supplementation significantly reduced CAC progression compared to control supplementation in a subgroup of participants who were ≥ 85% adherent to the intervention [80]. In another RCT including 42 nondialyzed patients with CKD stages 3–5 for 270 days, the increase of common carotid intima-media thickness (CCA-IMT) was significantly lower in the supplementation group with MK-7 at a dose of 90 μg together with 10 μg of cholecalciferol (vitamin D) (K + D group) compared with the D group [81]. The increase of CAC was slightly lower in the K + D group than in the D group, but the difference was not significant [81]. In a single arm study of 26 patients supplemented with 45 mg MK-4 for 1 year, the annual increase of CAC was 14%, but brachial ankle pulse wave velocity was not significantly changed [82]. In a recent RCT among 68 patients with type 2 diabetes and CVD for 6 months, MK-7 supplementation (360 μg/day) did not significantly change calcification mass score of both femoral arteries compared with placebo treatment [83]. RCTs designed to assess the effect of PK and MK-7 on CAC in HD patients are ongoing [84,85,86].

### 6.3. Vitamin K Antagonists and CAC

Vitamin K antagonists (VKAs) including coumarins (warfarin, acenocoumarol, and phenprocoumon, etc.) and indandiones (fluindione, etc.) are oral anticoagulants utilized for the treatment of thromboembolic diseases [87]. VKAs exert their anticoagulant effect by interfering with vitamin K cycle [88]. VKAs inhibit the action of VKOR to deplete the reduced (hydroquinone) form of vitamin K, thereby suppressing hepatic γ-carboxylation of VKDPs such as coagulation factors II, VII, IX, and X. In addition to the inhibitory effect on coagulation factors in the liver, warfarin also affects peripheral γ-carboxylation of VKDPs including MGP in vascular tissues [4]. Blocking γ-carboxylation of MGP in VSMCs by warfarin has been shown to contribute to its inhibitory effect on VSMC calcification [89]. Experimental studies using animal models have demonstrated that warfarin induces arterial calcification. In a rat model, warfarin caused acute medial calcification of major arteries and markedly increased the expression levels of MGP mRNA and protein in calcified arteries and decreased serum levels of MGP [90]. ApoE-deficient mice treated with warfarin developed atherosclerotic plaque calcification associated with features of plaque vulnerability [91]. Moreover, high doses of vitamin K reversed warfarin-induced medial vascular calcification in rats [92]. 

There are several clinical studies investigating the pro-calcific effect of warfarin on arterial calcification. In a cross-sectional study including 133 VKA users and 133 age, gender and the Framingham Risk Score (FRS) matched non-VKA users, the mean coronary calcification score (Agatston score) and the fraction of calcified coronary plaques significantly increased according increasing the duration of drug use in VKA users compared with non-VKA users [91]. In another cross-sectional study of 236 patients, use of VKA was significantly associated with the presence of ascending and descending aorta calcification compared with no anticoagulation treatment [93]. In a retrospective matched cohort study, a higher prevalence of lower extremity arterial calcification was found in patients receiving warfarin compared with those with no treatment [94]. The prevalence of breast arterial calcification was greater in women currently or in the past treated with warfarin compared with those without warfarin treatment [95]. The long-term effect of VKA use on coronary artery calcification has been investigated in 43 patients receiving VKA with metallic prosthetic valves and 65 control patients. CAC score assessed by CT was significantly higher in long-term VKA users compared with the control group [96]. In serial coronary intravascular ultrasound (IVUS) examinations performed during an 18- to 24-month period, a post hoc analysis of 8 prospective trials revealed that warfarin therapy was associated with progressive coronary plaque calcification independent of changes in atheroma volume [97]. 

## 7. dp-ucMGP as a Marker of Vitamin K Status

Active MGP is formed through sequential post-translational modification processes [50]. While carboxylation affords the capacity to bind calcium ions, phosphorylation may facilitate the cellular secretion process of MGP. After translation in the endoplasmic reticulum, MGP is firstly carboxylated by the action of GGCX requiring vitamin K as a cofactor, forming dephospho-carboxylated MGP (dp-cMGP). Next, a Golgi casein kinase phosphorylates dp-cMGP to p-cMGP (phosphorylated-carboxylated MGP), thereby facilitating secretion. p-cMGP is secreted into the extracellular space where it acts as an inhibitor of tissue calcification. Inactive dp-ucMGP is also released from cells into the bloodstream. Subsequently, three forms of MGP, that is, dp-ucMGP, dp-cMGP, and p-cMGP are present in the circulation. Vitamin K deficiency may impair the activity of MGP. Among these three conformations, dp-ucMGP is recognized as the best single biomarker of vitamin K deficiency [98]. In the general population, circulating levels of dp-ucMGP increase with age and with deterioration of renal function [50]. 

There are several studies to investigate whether vitamin K supplementation could decrease plasma levels of dp-ucMGP and subsequently reduce the risk of CV events. In a randomized, double-blind, placebo-controlled trial of 60 participants from the general population for 12 weeks, plasma levels of dp-ucMGP were dose-dependently decreased in the 180 μg and 360 μg menaquinone-7 (MK-7) supplementation groups by 31% and 46%, respectively, whereas dp-ucMGP levels remained unchanged in the placebo group [99]. In another double-blind, placebo-controlled trial of 244 healthy postmenopausal women for 3 years, MK-7 supplementation reduced dp-ucMGP levels by 50% in comparison with the placebo group [100]. A prospective, randomized, single-blind intervention study conducted with 200 chronic HD patients for 8 weeks showed that MK-7 supplementation dose-dependently decreased dp-ucMGP levels in the groups with 360, 720, 1080 μg thrice weekly by 17, 33, and 46%, respectively [101]. These data suggest that MK-7 supplementation may reduce the circulating levels of dp-ucMGP through enhancement of MGP activation. Therefore, it is likely that vitamin K2 intake might be beneficial for prevention of future CV events. 

## 8. dp-ucMGP as a Novel Marker of VC

The association between blood levels of MGP including its inactive forms and vascular calcification has been investigated in various populations [102,103,104,105,106,107,108,109]. In a prospective cohort study of 571 women aged 57.3 years, plasma levels of dp-ucMGP were borderline significantly associated with the presence of CAC [106]. A cross-sectional study conducted with 198 type 2 diabetes mellitus (DM) patients with normal or slightly impaired kidney function revealed that circulating dp-ucMGP was independently associated with below-knee arterial calcification score [107]. Plasma levels of dp-ucMGP were positively associated with the aortic calcification score in a cohort consisting of 107 patients with CKD stages 2–5 [108]. An observational cohort study of 136 hemodialysis (HD) patients indicated that plasma levels of dp-ucMGP were significantly associated with the vascular calcification score [109]. These studies on various populations, including healthy subjects and patients with high risks of CV, suggest that plasma dp-ucMGP may be a useful marker of VC. 

## 9. The Association of dp-ucMGP with CV Events and Mortality

Vascular calcification is independently associated with an increased risk of CV events and mortality [110,111,112,113,114,115,116]. Since there are accumulating evidence suggesting that plasma dp-ucMGP is a useful marker of VC, several investigators have explored the possible association of dp-ucMGP with CV events and mortality in various populations. In a study among 577 older adults of the Longitudinal Aging Study Amsterdam study (LASA) with no history of previous CVD, there was a more than 2-fold higher risk of CVD in the highest tertile of dp-ucMGP group (HR: 2.69, 95% CI: 1.09–6.62) compared with the lowest tertile after a follow-up period of 5.6 years [111]. In a Flemish population study of 2318 participants, higher concentrations of dp-ucMGP were an independent predictor of total, non-cancer, CV mortality after a follow-up period of 14.1 years [115]. In a prospective cohort study consisting of 518 type 2 DM patients, high dp-ucMGP levels were associated with increased risk for CVD, especially with peripheral artery disease (PAD) and heart failure after a follow-up of 11.2 years [110]. In a prospective cohort study conducted with 799 patients with history of myocardial infarction, stroke, or coronary artery disease, there was a higher risk of all-cause and CV mortality in the highest quartile of dp-ucMGP (HR 1.89, 95% CI: 1.32–2.72 and HR 1.88, 95% CI: 1.22–2.90, respectively) [116]. These data suggest that plasma dp-ucMGP may be a novel biomarker for CV events and mortality.

## 10. Conclusion

Vascular calcification is an inflammation-mediated process in which activated transdifferentiation of VSMCs into osteoblastic cells was induced by the action of cytokines such as TNF-α and OSM secreted from macrophages infiltrated in intimal and medial tissues of arterial wall. Vitamin K may prevent vascular calcification through an anti-inflammatory mechanism as well as promotion of γ-carboxylation of MGP. The preventive role of active MGP in vascular calcification has been proven by several clinical studies demonstrating the procalcific effect of warfarin, a vitamin K antagonist in arteries. Various clinical studies have revealed that higher vitamin K intake may reduce the risk of vascular calcification and CVD and higher plasma concentrations of dp-ucMGP may predict future risk of death or CV events. High-quality prospective cohort studies and RCTs are still required to establish the role of vitamin K in CV health.

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
