# Peer review of "The Inhibitory Roles of Vitamin K in Progression of Vascular Calcification"

_nutrients, 2020, doi:10.3390/nu12020583_

Round 1

Reviewer 1 Report

The study of Shioi et al presents a clear and timely review on the connection between the process of vascular calcification and vitamin K and/or vitamin K-dependent proteins (VKDPs). The review is well written and it references most of the relevant studies in the literature.

My main criticisms regards the lack of a section on the inflammatory regulatory functions of several vitamin K-dependent proteins (VKDPs) that could affect the calcification process. Although the main regulator of calcification is matrix-Gla protein (MGP), as explained in the review, several additional VKDPs exert modulatory effects on inflammation, including Gas6, protein S and protein C. All these proteins interact with membrane receptors expressed in endothelial cells, inflammatory cells and VSMCs and have been implicated or studied in the calcification process. There are several studies on the function of these proteins on the process of vascular calcification, both in vitro and in vivo. These aspects should be commented in the introduction of the revision, and I suggest to provide a detailed account of these studies in a separate section of the text.

In the other hand, I do not understand the focus on oncostatin M (OSM). The implication of OSM in the process of calcification is described in detail (Line 16 of the abstract and Section 3.). This is confusing, as OSM is not a VKDP and there is no direct interaction of vitamin K with OSM.  I suggest to remove this part of the text.

The authors put a strong point on the “direct” anti-inflammatory effect of vitamin K (introduction and section 4). However, although studies in cell cultures have documented these effect, I am not certain that it is clearly established as a function in vivo. For instance, reference 11 (Line 66) is not indicative of a direct anti-inflammatory effect of vitamin K, rather of a correlation of vitamin K intake and CVD.

A graphical abstract could help the reader.

Minor points:

Line 62. Correct the grammar.

Paragraph lines 69-78. Apart from the indicated functions, several VKDPs exert modulatory effects on inflammation, including Gas6, protein S and protein C. These aspects should be commented in the introduction of the revision, even if it is not a main point of the revision.

Line 95. It is most commonly used the expression “oxidative stress”.

Section 4. VKDPs with anti-inflammatory functions (see previous comment).

L185. Also the rest of vitamin K antagonists, as coumarins (including warfarin, acenocpoumarol and dicoumarol, among the most commonly used drugs) and indandiones.

Author Response

The comments of the reviewer have been helpful in allowing us to revise our manuscript.  Regarding the comments suggested by the reviewer, we have attempted to respond as follows:

1) The reviewer's main criticisms regarding the lack of a section on the inflammatory regulatory functions of several vitamin K-dependent proteins (VKDPs) that could affect the calcification process

We have added the new paragraphs describing the anti-inflammatory functions of protein S and protein C  in the section 1 (Introduction) and the roles of Gas6, GRP, protein S, and protein C in vascular calcification in the section 5 (MGP and other VKDPs in Vascular Calcification), respectively.  

2) The reviewer's suggestion: removing the part of text describing the implication of OSM in vascular calcification

We have deleted the section describing the roles of OSM in vascular calcification and heterotopic ossification and added the roles of NF-kB in osteogenic differentiation of VSMCs and vascular calcification in the section 3 (Inflammation and Vascular Calcification).

3) The reviewer's criticism regarding poor in vivo evidences of anti-inflammatory roles of vitamin K

We have added several in vivo evidences suggesting the anti-inflammatory roles of vitamin K in the section 3 (Inflammation and Vascular Calcification).  

4) A graphical abstract could help the reader. 

Since the detailed mechanisms of vitamin K inhibitory action on vascular calcification remain unclear, it seems difficult to describe graphically the roles of vitamin K in vascular calcification. 

Minor points:

1) We have corrected the sentence as the reviewer pointed out.

2) We have added the new paragraphs regarding VKDPs such as Gas6, protein S and protein C as mentioned above.  

3) We have changed the expression as "oxidative stress".

4) According to the reviewer's suggestion, we have made the subsection titled as "Vitamin K Antagonists and CAC" and added the comments on other  vitamin K antagonists such as coumarins and indandiones.  

Reviewer 2 Report

The subject discussed in the paper is very important and precise extensive discussion. This review contains information of interest, but focuses primarily on indirect aspects related to the activity of Vitamin K in pathological calcification processes. The authors indicate that there are several clinical trials demonstrating that a high consumption of vitamin K reduces the risk of cardiovascular calcification. Nevertheless, these studies are explained very poor and superficially. In this regard, these studies should be discussed exclusively and much more extensively in a specific section (Vitamin K and CAC). Therefore, it is recommended to include a section where only studies in which the increase in Vitamin K consumption clearly demonstrated a decrease on the risk of cardiovascular calcification (these studies should be compared and discussed in depth).

Author Response

The comments of the reviewer have been helpful in allowing us to revise our manuscript. Regarding the comments suggested by the reviewer, we have attempted to respond as follows:

According to the reviewer's suggestion, we have separated the section (Vitamin K and CAC) into three subsections (1. Vitamin K status and CAC; 2. Vitamin K supplementation and CAC; 3. Vitamin K antagonists and CAC) and added more references in each subsection and discuss more extensively.  

Reviewer 3 Report

This is a nice, comprehensive review on the role of vitamin K in vascular calcification. The paper does not bring much new information, though.

On page 3 and 4 the authors comment on the anti-inflammatory effects of vitamin K; they state that it is likely that vitamin K may prevent inflammatory vascular disease. This statement should be mitigated because the observational data do not exclude the possibility that the two phenomena are independent from each other. It is also conceivable that a fall in inflammatory variables is an epiphenomenon.

Since this paper deals with the inhibitory role of vitamin K in the progression of vascular calcification, it would be helpful to rearrange the paper in such a way that we have a heading on what happens in case of K-deficiency and a heading on K-supplementation. The section on clinical studies in patients with warfarin or comparable agents should be expanded (there are more observations).

Author Response

The comments of the reviewer have been helpful in allowing us to revise our manuscript.  Regarding the comments suggested by the reviewer, we have attempted to respond as follows:

1) Regarding the reviewer's criticism of poor evidences supporting the anti-inflammatory effects of vitamin K on inflammatory vascular disease including calcification, we have basically agreed with the reviewer.  In order to reinforce our hypothesis, we have added the additional in vitro and in vivo evidences in the section 4 (Anti-inflammatory effects of vitamin K) in the revised manuscript.  

2) According to the reviewer's comment, we have separated the section 6 (Vitamin K and CAC) into three subsections (1. Vitamin K status and CAC; 2. Vitamin K supplementation and CAC; 3. Vitamin K antagonists and CAC) and added more references and discussed extensively in the revised manuscript.  

Round 2

Reviewer 2 Report

The paper has been widely modified and the current version can be published.